**Data Availability Statement:** All relevant data are within the paper and its Supporting Information files.

# Economic model to examine the cost-effectiveness of FlowOx home therapy compared to standard care in patients with peripheral artery disease

**Victory 'Segun Ezeofor**[1]*, **Nathan Bray**[1], **Lucy Bryning**[1], **Farina Hashmi**[2], **Henrik Hoel**[3,4], **Daniel Parker**[2], **Rhiannon Tudor Edwards**[1]

1 Centre for Health Economics and Medicines Evaluation, Bangor University, Bangor, Gwynedd, Wales, United Kingdom, 2 School of Health and Society, University of Salford, Manchester, United Kingdom, 3 Otivio AS, Oslo, Norway, 4 Department of Vascular Surgery, Oslo University Hospital, Oslo, Norway

* v.ezeofor@bangor.ac.uk

## Abstract

### Background

Critical limb ischaemia is a severe stage of lower limb peripheral artery disease which can lead to tissue loss, gangrene, amputation and death. FlowOx™ therapy is a novel negative-pressure chamber system intended for home use to increase blood flow, reduce pain and improve wound healing for patients with peripheral artery disease and critical limb ischaemia.

### Methods

A Markov model was constructed to assess the relative cost-effectiveness of FlowOx™ therapy compared to standard care in lower limb peripheral artery disease patients with intermittent claudication or critical limb ischaemia. The model used data from two European trials of FlowOx™ therapy and published evidence on disease progression. From an NHS analysis perspective, various FlowOx™ therapy scenarios were modelled by adjusting the dose of FlowOx™ therapy and the amount of other care received alongside FlowOx™ therapy, in comparison to standard care.

### Results

In the base case analysis, consisting of FlowOx™ therapy plus nominal care, the cost estimates were £12,704 for a single dose of FlowOx™ therapy per annum as compared with £15,523 for standard care. FlowOx™ therapy patients gained 0.27 additional quality adjusted life years compared to standard care patients. This equated to a dominant incremental cost-effectiveness ratio per QALY gained. At the NICE threshold WTP of £20,000 and £30,000 per QALY gained, FlowOx™ therapy in addition to standard care had a 0.80 and 1.00 probability of being cost-effectiveness respectively.

**Funding:** This paper presents independent research funded by the European Commission, through the Horizon 2020 Fast Track to Innovation funding scheme. (Reference Number 737964). The views expressed are those of the author(s) and not necessarily those of the funding body. The funder (Otivio AS) who owns the commercial rights to the FlowOx product, received a grant from the Norwegian Research Council (NFR grant no: 285758) and thereby provided support in the form of salaries for author HH. Otivio AS did not have any role in the study design, data collection and analysis, decision to publish, or preparation of the manuscript. The specific roles of these authors are articulated in the 'author contributions' section. Otivio AS provided information about the cost of the FlowOx device.

**Competing interests:** HH is employed by Otivio AS with funding from The Research Council of Norway (grant no. 285758). Otivio AS has the commercial rights to the INP technology used in this study. The authors alone are responsible for the content and writing of the paper. We adhere fully to PLOS ONE policies on sharing data and materials.

## Conclusions

FlowOx™ therapy delivered as a single annual dose may be a cost-effective treatment for peripheral artery disease. FlowOx™ therapy improved health outcomes and reduced treatment costs in this modelled cohort. The effectiveness and cost-effectiveness of FlowOx™ therapy is susceptible to disease severity, adherence, dose and treatment cost. Research assessing the impact of FlowOx™ therapy on NHS resource use is needed in order to provide a definitive economic evaluation.

## Introduction

Peripheral artery disease (PAD) is estimated to affect around 13% of adults aged over 50 years old in Western populations [1]. Lower limb PAD is a manifestation of systemic atherosclerotic disease and is commonly caused by hardening of the arteries, where blood flow to affected limbs is restricted due to obstruction of atherosclerotic plaque. While many patients are asymptomatic, around 5% of older adults experience symptomatic PAD with intermittent claudication (IC), causing pain on walking [1].

PAD is associated with an increase in morbidity, reduction in mobility and impaired quality of life (QoL). In its severe forms, such as critical limb ischaemia (CLI), PAD can lead to tissue loss (i.e. ulceration), gangrene and lower limb amputation (LLA). The classification of PAD varies from the mild asymptomatic stage where there is a partial artery obstruction to severe necrosis and/or gangrene of the limb, leading to amputation [2].

The population of people affected by PAD is large and the condition has a long time period of morbidity. This has a significant impact on the prioritisation and allocation of health and social care resources [3]. With increasing life expectancy, high rates of diabetes and historically high rates of smoking, the prevalence of PAD is increasing globally [4,5]. In the first year after PAD diagnosis, there is a substantial increase in healthcare costs related to PAD follow-up and lower limb related procedures [6].

Between 1–3% of people experiencing IC develop limb-threatening complications such as CLI within 5 years [7]. Outcomes for patients with CLI are poor, with overall survival rates considered worse than for many types of cancer [7] though these rates could be impacted by related co-morbidity [5]. Where revascularization is not possible, patients with CLI have a one year mortality rate of 25% [8,9], rising to 50% by five years [7].

CLI is typically characterised by a range of clinical indicators and complications, including severely diminished circulation, ischaemic pain, ulceration, tissue loss, and even gangrene [10]. Patients with non-reconstructable CLI, where the limb cannot be restored by treatments such as angioplasty or bypass surgery may be offered amputation [8]. Between 25–40% of patients with CLI require major amputation within the first 12 months after diagnosis [2,7]. Approximately 5000 major LLA procedures are performed in the UK each year [4]. For diabetic foot ulcers and amputation alone, the healthcare cost in 2014–2015 was estimated at between £837 million and £962 million [11]. It is generally accepted that lower extremity amputations have particularly poor outcomes with regards to health-related QoL, mortality and medical costs [4,12]. For example, mortality rates for patients with CLI are high following amputation, with a 35% probability of mortality in the first year and a 19% probability every year thereafter [8].

In research literature, there is little consensus on the definition of lower extremity amputation and severity levels (e.g. minor, major, all) [13]. NICE guidance defines that amputations

for PAD above the ankle level are considered 'major', with further categorisation of levels of amputation as follows [8]:

- Toe: one or more toes is removed, often with the metatarsal heads

- Transmetatarsal: all of the toes removed together with the metatarsal heads

- Trans-tibial: the leg is removed about a hands-breadth below the knee; known as below knee amputation

- Trans-femoral: the leg is removed about a hands-breadth above the knee; known as above knee amputation

The health burden of PAD, CLI and related complications has led to an increased focus on non-invasive therapeutic interventions aimed at improving the clinical signs/symptoms and wound healing, especially in a cohort that is often deemed unable to benefit from surgery. Even the use of the most effective pharmacologic agents results in only a modest improvement of symptoms, such as IC and rest pain. Although drug therapy, control of risk factors and revascularisation procedures can prove beneficial to patients suffering with CLI, for many patients the symptoms continue to deteriorate leading to loss of tissue viability and an increased risk of amputation.

The latest NICE guidance states that due to rapid changes in diagnostic methods, endovascular treatments, and vascular services, there is a great deal of uncertainty about the best way to manage patients with PAD, which in turn has led to variance in services across the NHS [8,10]. CLI and lower limb ulcers place an enormous burden on the individual patient and informal carers, and have substantial financial implications on health and social care systems [14]. Outpatient dressings and nursing time comprise the majority of costs associated with the care of patients with ulcers [15]. Jeffcoate and Harding [16] argue that for patients with foot ulcers potentially leading to amputation, interventions should be directed at infection, peripheral ischaemia, and abnormal pressure loading caused by peripheral neuropathy and limited joint mobility.

The development of home-use devices that apply intermittent pneumatic foot and/or leg compression have been shown to improve arterial flow to the lower limb [17]. Pilot data on the effects of negative pressure therapy has shown promising results with regard to improving haemodynamic outcomes and ischaemic ulcer healing in patients with PAD [18,19]. Clinical studies conducted on a few patients using the FlowOx$^{TM}$ therapy in Norway showed some signs of positive result [51,52]. However, at present there is limited evidence of the cost-effectiveness of different pressure-based therapies, such as FlowOx™ therapy, to improve vascular lower limb health.

The FlowOx™ (Otivio, Oslo, Norway) therapeutic device is a CE marked medical device (Certificate Number: 247096-2017-CE-NOR-NA-PS Rev. 2.0). The therapy is intended for use by patients who present with reduced blood flow (ABPI <0.8) or symptoms of claudication in advance of surgical intervention, but has also been used for treatment of patients with critical limb ischaemia post intervention. The device is likely to have its biggest impact in patients who are in a stable but critical condition with few treatment alternatives other than palliative care, chronic wound care or amputation. The device comprises a wearable boot, inflatable padding and seal which together act as a chamber, allowing an intermittent negative pressure to be applied to the leg below the knee (Fig 1). It is intended to be used in a clinical or domestic environment. The padding and seal are designed to ensure that the limb is suspended within the boot (therefore preventing pressure points between the skin and the boot) and that an adequate vacuum can form once the negative pressure is applied. The boot is connected to a

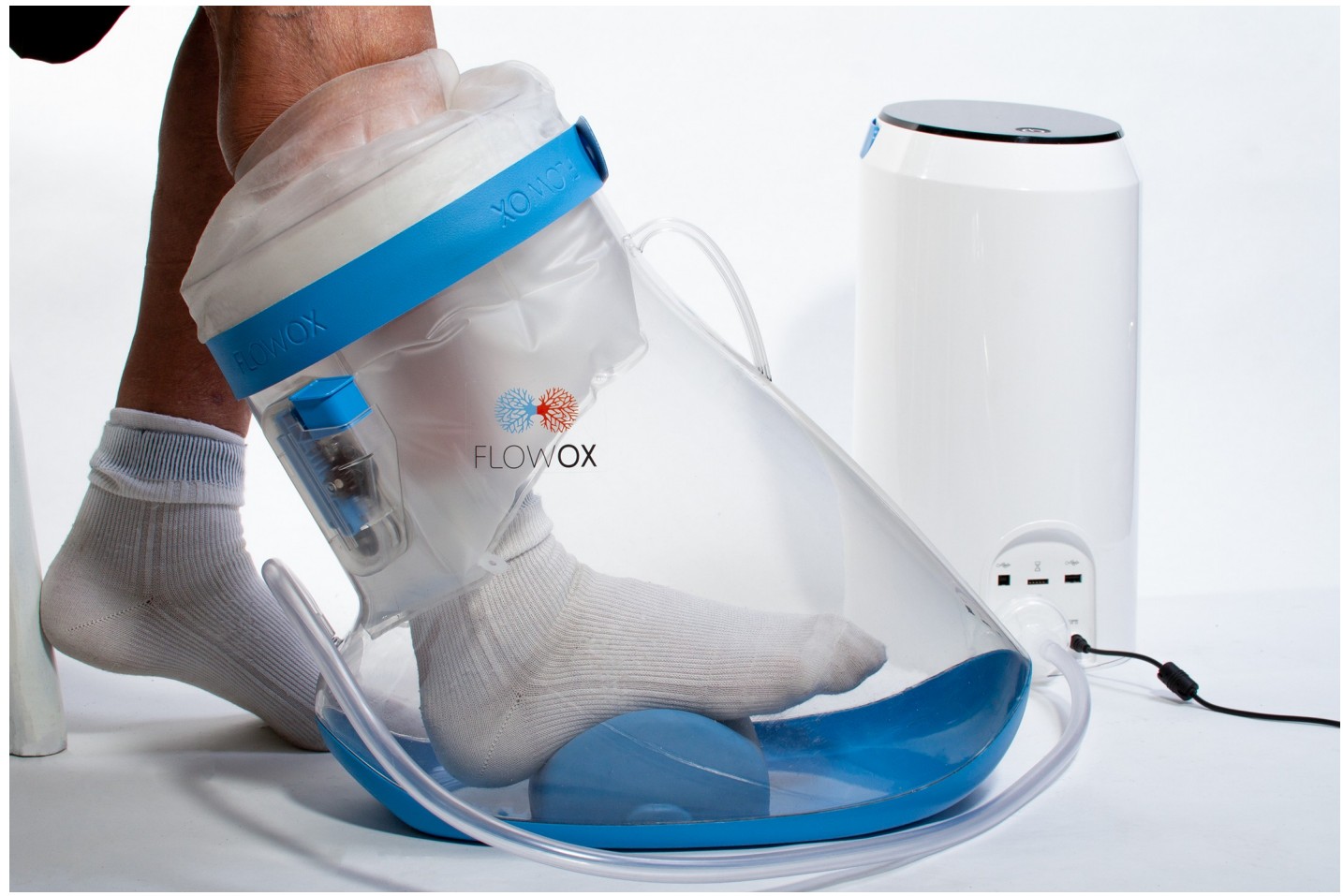

**Fig 1. FlowOx™ devices and components.**

control unit which generates a pulsed 'on-off' negative pressure (-40 mmHg and ambient pressure for 10 sec and 7 sec, respectively). The pressure cycle is pre-set and cannot be changed by the user. Patient use data is recorded directly by the control unit.

As part of an international European study to assess the effectiveness of FlowOx™ therapy compared to standard care in lower limb PAD patients (including IC, CLI and lower leg ulcers), we developed of an economic model to explore the cost-effectiveness of FlowOx™ therapy in a modelled cohort of patients with PAD. The project was funded by the European Commission, through the Horizon 2020 Fast Track to Innovation funding scheme (737964).

## Methods

### Model overview and rationale

From an NHS perspective, we developed a Markov model to estimate the cost-effectiveness of FlowOx™ therapy in treating patients with PAD. We used a range of methods to synthesise evidence for the economic model.

Data used to populate the model were sourced from the UK based LLIFT (Lower Limb Ischaemia FlowOx™ Therapy) pilot trial (ISRCTN51433523), supplemented with additional data from a Norway based double-blind randomised controlled trial (NCT03640676) and

other existing published literature. The model was used to generate cost per quality-adjusted life year (QALY) estimates for FlowOx™ therapy compared to standard NHS care. Our original intention was to exclusively use primary data from the LLIFT trial, which focused on a patient population with severe stage PAD, however due to the limited sample size and limited data from the LLIFT trial it was necessary to widen out the data sources in order to sufficiently populate the model. We therefore utilised supplementary data from the Norway trial, which focused on patients with mild stage PAD, and additional published evidence. A systematic review to identify published evidence was planned [20], however due to the extent and variability of literature identified on the topic it was considered outside the scope of this study to complete a full systematic review. Comprehensive searches of databases were conducted as described in the published review protocol [20] in order to identify a robust pool of evidence to conduct rapid scoping searches for each data source required in the model.

Scoping through published systematic reviews [21] and other relevant literature [22,23] helped to inform our approach to the economic evaluation, including the model type and structure. Our modelling approach was based on scenario analysis to offer a base case reference scenario with additional scenarios modelled as part of the sensitivity analysis.

## Decision model

The decision problem facing health care commissioners in this context in the UK or internationally is whether to invest in and provide to PAD patients FlowOx™ therapy in order to help manage their condition. Our objective was to use an economic model to calculate the relative cost-effectiveness of the FlowOx™ therapy as part of a pathway of care as compared with standard NHS care (without the FlowOx™ therapy) in a hypothetical cohort of patients with mild to severe PAD. Although FlowOx$^{TM}$ therapy is intended for use by patients who present with reduced blood flow (ABPI <0.8) or symptoms of claudication in advance of surgical intervention, the device may also provide some preventative options, thus we conducted a cost analysis for all disease states in order to evaluate the most appropriate and cost-effective stage at which to begin therapy.

In constructing our model, we opted for a Markov model, with three FlowOx™ therapy scenarios:

1. The FlowOx™ therapy plus nominal care (S0)

2. The FlowOx™ therapy only (S1)

3. The FlowOx™ therapy plus standard care (S2).

We chose the base case analysis as FlowOx™ therapy plus nominal care, and standard care as the comparator. We chose FlowOx™ therapy plus nominal care as the base case to examine the impact of substituting FlowOx™ therapy for standard care. The inclusion of nominal care was to acknowledge that patients would still need regular monitoring and clinical input. We defined nominal care is a reduced standard care, comprising fewer nurse visits and reduced medication use. Costs associated with different care options can be found in Table A of S1 Appendix; while costs associated with different health states can be found in Tables B and E of S1 Appendix. See Table 1 for an overview of study intervention scenarios.

**Rationale for the model structure.** Our model was designed to mimic the progression of the PAD to lower limb ulcers and CLI. To allow for this flexibility, a Markov model was developed. Each state in the Markov model represents a plausible, discrete health state. We used transition probabilities to describe the causal relationships and the probability of a patient moving from one state to another. Sources of data used in the model are shown in Tables A, B and E in S1 Appendix.

**Table 1. Intervention scenario definitions for Markov model.**

| Scenario | Definition |
|---|---|
| Standard care (SC) | Rest pain managed with appropriate analgesics in accordance with the NICE guidance on the treatment of peripheral arterial disease [8,10] (see Tables A,B and E in S1 Appendix for cost). |
| Scenario 0 (S0) FlowOx™ therapy plus nominal care | FlowOx™ therapy is delivered as a partial substitution to standard care as a self-administered home based therapy with self-monitoring. Nominal medical attention also received, consisting of limited specialist nurse visits (see Table A in S1 Appendix for cost) to support use of the FlowOx™ device. The level of nominal care is dependent on the severity of PAD. |
| Scenario 1 (S1) FlowOx™ therapy alone | FlowOx™ therapy is delivered as a complete substitution to standard care as a self-administered home based therapy with self-monitoring. No additional medical attention or support to use the device. |
| Scenario 2 (S2) FlowOx™ therapy plus standard care | FlowOx™ therapy is delivered in addition to all standard care treatment, with no substitution of care. |

Markov modelling allows complex real-life events to be represented in a simplified health state form [24]. In our Markov model we followed a cohort of 1000 patients moving through defined Markov states and time periods. The modelled patients can remain in their current health state, move to another health state or reach the 'absorbed' state (a health state with no cost or health benefit e.g. dead in this model) according to certain transition probabilities [25]. The Markov model had finite and independent states with each state being assigned a health utility and cost value to allow cost per QALY estimates to be calculated. Parameters such as clinical outcomes, healthcare costs incurred, time duration of the intervention and quantity of intervention were included in the model construction.

Our aim was to create a realistic but simplified model, which took account of the complexity of the issue, whilst also allowing ease of modelling. The challenge of modelling the cost-effectiveness of diseases such as PAD is that they have asymptotic and symptomatic levels, and each level has sub-levels. In clinical practice, the Rutherford and Fontaine classification are used to distinguish between these different levels [26,27].

**Disease states/pathways.** Our Markov model states, and transition probabilities describe, to the best of our ability, the biological progression of disease severity of PAD, LLI, lower limb ulcers and CLI. The Fontaine and Rutherford classifications, as shown in Table 2, were applied in the criteria and boundary requirements for the Markov state classification. We used these

**Table 2. State classification of lower limb PAD severity.**

| State | Definition | Rutherford | Fontaine | ABI* |
|---|---|---|---|---|
| Mild | Asymptomatic | Stage 0 | Stage I | ABI > 0.95 |
| Progressive (Prog) | Mild Claudication | Stage 1 | Stage IIA (>200m walking) | ABI > 0.80 |
| | Moderate Claudication | Stage 2 | Stage IIB (< 200m) | ABI > 0.40 |
| | Severe Claudication | Stage 3 | | |
| | Rest pain | Stage 4 | Stage III | ABI < 0.40 |
| Severe | Ischemic ulceration (limited to digits) | Stage 5 | Stage IV | TP <30mm Hg |
| | Severe ischemic ulceration or frank gangrene | Stage 6 | | |
| Minor Amputation (MinAmp) | Amputation of the digits up to the ankle. | N/A | N/A | N/A |
| Major Amputation (MajAmp) | Amputation above the ankle. (a) Below the Knee (b) Above the Knee | N/A | N/A | N/A |
| Recovered | Recovered from lower limb ischemia after amputation | N/A | N/A | N/A |
| Dead | No Longer alive | N/A | N/A | N/A |

*Ankle Brachial (Pressure) Index (ABPI/ABI): The ratio of blood pressure at the ankle to the pressure at the upper arm [31].

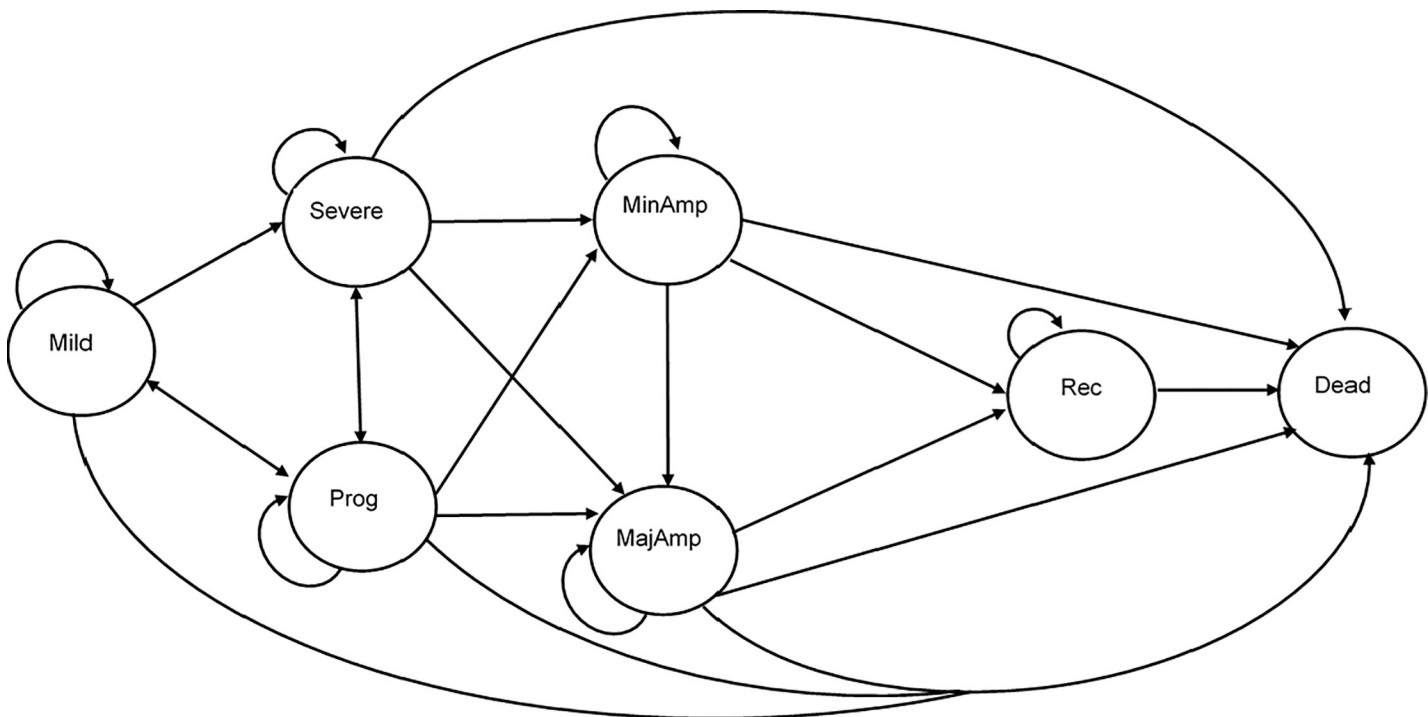

**Fig 2. A simplified model diagram of the probable patient transition between health states.**

classifications to investigate progression through health states (see Fig 2 for a pictorial representation of the model structure). The model consists of seven-disease states: mild, progressive, severe, minor amputation, major amputation, recovered and a dead state (which acts as the absorbed state). The recovered state is dependent on whether the patient has undergone at least one amputation or not; this state has a health index less than the 'mild' state due to the presence of disability. See Table C in S1 Appendix for utilities for each health state. The structure of the Markov model includes the amputation states as different severity stages due to the impact on patient outcomes [28–30].

**Cycle length.** For a robust and valid comparison of the cost-effectiveness of FlowOx™ therapy, the cycle length in this Markov model was three months, this was constructed to match one complete dosage of FlowOx™ therapy. The cycle length was intended to be short enough to capture patient transition and long enough to include all health benefits observed.

**Time horizon.** A time horizon of five years was adopted for this health economic model analysis to allow for ample time to reflect the trajectory towards costs and outcomes between patients receiving FlowOx™ therapy and patients receiving standard care. The time horizon allows time for many of the patients in more severe stages of PAD, for example patients experiencing CLI, to progress through the model structure with rates of major amputation within the first year estimated to be up to 40% [2] and mortality estimated as high as 35% within one year and 50% within five years [8].

**Analysis.** The two FlowOx™ therapy trials described earlier had a follow up of three months, the observed probabilities from the clinical follow up and published evidence was used to populate the transition probabilities. Where there was insufficient data, point estimates were derived from published literature [3,32]. The relative risk and effects of disease progression were derived from the study data and are presented with the transition probabilities in Table D in S1 Appendix.

The model assumed all participants were at the symptomatic phase of PAD (progressive state) at the beginning of the model and then progressed through the model using the given transition probabilities. Cost and health outcomes were computed from the distributed simulation.

**Sources of cost data.** The sample sizes of the primary data sources were limited, hence point estimates from similar studies were combined to give a more robust model. Cost data associated with PAD have a wide variance, which has a direct impact on average cost estimates. Costs were based on the average of unit cost in the UK NHS, for 2017/18, and were expressed in pounds Sterling (£GBP). The cost estimates for the Markov states and patient cohort were pooled from the national schedule of NHS reference costs [33] and other published literature [11,34,35]. These costs were inflated in accordance to the Hospital and Community Health Services pay and price index (HCHS) [36]. Resource use costs included community, inpatient and outpatient costs for healthcare service use. The range of unit costs used for each model state are presented in Table E in S1 Appendix. An NHS perspective was adopted in this study, as there was insufficient evidence to allow a wider, societal perspective. The Markov state costs by level of severity are summarised in Table 3.

These annual costs were rescaled to match the cycle length (i.e. three months) of the model (see Table E in S1 Appendix). Where these costs had different prices, the mean was adopted and these costs were discussed and agreed upon by expert advisors. Expert advisors comprised a vascular surgeon and nurses working within secondary care vascular medicine. This model focused on FlowOx™ therapy using a rental-based pricing structure, at a cost of £15 per day to NHS services, as advised by the manufacturer. No overheads were added to the FlowOx™ therapy due to the predominant focus on patient-led home use.

## Source of health-related QoL outcomes

We measured effectiveness of the FlowOx™ device in terms of its relative impact on health status, measured in terms of QALYs gained. QALYs were calculated using EQ-5D-5L data from the primary data sources and published literature [37].

The EQ-5D-5L measures health-related QoL in five dimensions (mobility, self-care, usual activities, pain-discomfort, and anxiety-depression) each with a five-point scale of severity [38,39].

The EQ-5D-5L data from the primary data sources was used to generate health-related QoL outcomes and utility data for participants in the intervention programme. Using this data and other published evidence [37] we developed QALY estimates as the primary health outcome measure. An estimated health state utility value is assigned to each health stage in the model. We calculated QALYs gained using the area under the curve method [40]. To avoid bias we adjusted for differences in baseline EQ-5D-5L scores [41,42].

**Table 3. The Markov state mean cost per cycle, by level of model state severity.**

| State | Mean cost (£) per three-month cycle |
|---|---|
| Mild | 143.09 |
| Progressive | 828.43 |
| Severe | 1767.33 |
| Minor Amputation | 2254.98 |
| Major Amputation | 4913.19 |

Note: Costs are reported in 2017/2018 financial year and were inflated in the computation to 2018/19. Recover state does not have PAD treatment costs.

## Discounting

In accordance with NICE guidance [43], our analysis applied a discount rate for cost at 3.5% and a discount rate for effect at 3.5%, considering the potential for FlowOx™ therapy to be a preventative therapeutic intervention. As a sensitivity analysis we varied the discount rate on effect from 0% to 3.5% in accordance with NICE guidance on methods to evaluate medical technologies [44].

## Data analysis

Data included in the study considered individuals who had PAD following either the Fontaine classification or the Rutherford classification. The study data for the model are summarised in Tables 4 and 5. No formal sample size calculation was conducted prior to commencement of the original study due to the lack of existing data for efficacy of FlowOx™ therapy on lower limb ulcer healing in patients with CLI. We assessed the quality of our data by checking for accuracy, completeness and relevance. To prevent loss of precision, missing data were investigated for relevance and excluded from the analyses if there was significant risk of bias.

## Sensitivity analysis and data uncertainty

Both deterministic and probabilistic sensitivity analyses (PSA) were undertaken to check for robustness in the model. The PSA conducted was multivariate, capturing ranges of variables used in the model to explore uncertainty during the Monte Carlo simulation.

**Table 4. Demographics of study population (generated from two primary data sources).**

| Characteristics | FlowOx™ Intervention N = 35 | Control N = 27 |
|---|---|---|
| **Demographics** | | |
| Gender | | |
| Male (%) | 22 (62.86) | 22 (81.48) |
| Female (%) | 13 (37.14) | 5 (18.52) |
| Mean age (SD) | 71.23 (8.97) | 72.96 (6.33) |
| Mean height (SD) | 164.13 (28.97) | 174.85 (6.56) |
| Mean weight (SD) | 80.82 (13.11) | 83.90 (14.15) |
| **Rutherford Classification (%)** | | |
| Stage 0 | 0 | 0 |
| Stage 1 | 1 (2.86) | 0 |
| Stage 2 | 14 (40.00) | 19 (70.37) |
| Stage 3 | 4 (11.43) | 4 (14.81) |
| Stage 4 | 4 (11.43) | 0 |
| Stage 5 | 8 (22.86) | 3 (11.11) |
| Stage 6 | 0 | |
| **Amputation (%)** | | |
| Minor | 2 (5.71) | 1 (3.70) |
| Major | 2 (5.71) | 0 |
| **Minimum walking distance (%)** | | |
| ≤ 200 | 19 (55.88) | 19 (70.37) |
| > 200 | 2 (5.88) | 4 (14.82) |
| **Ulcer [From stage 4 -] (%)** | | |
| 1 | 7 (20.59) | 2 (7.41) |
| 2 | 1 (2.94) | 0 |
| ≥ 3 | 5 (14.71) | 2 (7.41) |

**Table 5. Baseline heterogeneity in health-related QoL (i.e. EQ-5D-5L).**

| Characteristics | Intervention | | | | Control | | | |
|---|---|---|---|---|---|---|---|---|
| | Baseline | | Follow-up | | Baseline | | Follow-up | |
| | Mean | SD | Mean | SD | Mean | SD | Mean | SD |
| **Gender** | | | | | | | | |
| Male | 0.5302 | 0.3151 | 0.5978 | 0.2420 | 0.6980 | 0.2279 | 0.6913 | 0.2705 |
| Female | 0.3968 | 0.3207 | 0.4567 | 0.3285 | 0.6226 | 0.1821 | 0.7663 | 0.3727 |
| **Age** | | | | | | | | |
| >70 | 0.5311 | 0.3192 | 0.5709 | 0.2890 | 0.7099 | 0.2656 | 0.7822 | 0.3256 |
| <70 | 0.3771 | 0.3099 | 0.4883 | 0.2767 | 0.6493 | 0.1055 | 0.5797 | 0.2130 |
| **Height** | | | | | | | | |
| >170 | 0.4891 | 0.3554 | 0.5802 | 0.2733 | 0.7035 | 0.2390 | 0.6818 | 0.2794 |
| <170 | 0.4649 | 0.3019 | 0.4941 | 0.3009 | 0.6710 | 0.1572 | 0.7586 | 0.1476 |
| **Weight** | | | | | | | | |
| >80 | 0.4375 | 0.3800 | 0.5954 | 0.2720 | 0.6695 | 0.1892 | 0.6576 | 0.2805 |
| <80 | 0.5093 | 0.2515 | 0.4589 | 0.2920 | 0.7031 | 0.1225 | 0.7685 | 0.2619 |
| **Rutherford Classification** | | | | | | | | |
| Stage 0 | - | - | - | - | - | - | - | - |
| Stage 1 | 0.5980 | 0.0000 | 0.4000 | 0.0000 | - | - | - | - |
| Stage 2 | 0.5524 | 0.3253 | 0.6789 | 0.1921 | 0.6799 | 0.1225 | 0.7419 | 0.2798 |
| Stage 3 | 0.5680 | 0.2474 | 0.6940 | 0.0581 | 0.7153 | 0.1080 | 0.6547 | 0.4225 |
| Stage 4 | 0.6648 | 0.0488 | 0.4310 | 0.2170 | - | - | - | - |
| Stage 5 | 0.4111 | 0.3021 | 0.5418 | 0.1928 | 0.6457 | 0.2346 | 0.5597 | 0.1324 |
| Stage 6 | - | - | - | - | - | - | - | - |
| **Amputation** | | | | | | | | |
| Minor | 0.2220 | 0.4370 | 0.2860 | 0.4257 | 0.7160 | 0.0000 | 0.7160 | 0.0000 |
| Major | -0.0870 | 0.0000 | -0.1115 | 0.2058 | - | - | - | - |

Where possible, distributions of data from the simulations were used to reflect parameter uncertainty. The probability of cost-effectiveness at different willingness to pay (WTP) thresholds were also investigated. Confidence intervals (CIs), developed using incremental monetary net benefit (IMNB), were calculated for number of values. Though it would be possible to develop many other model structures for this clinical area, our analysis utilised a biologically compact but flexible model structure which we feel was best suited to achieve our original aims. All necessary parameters were included with varying discount rates. We addressed heterogeneity by running our model 1000 times with different age sub-groups, different time intervals and different scenarios.

## Model validation assessment

To ensure good and standard practice the model was validated and verified in accordance with best practice guidance, hence we used the Assessment of the Validation Status of Health-Economic decision models (AdVISHE) tool [45] and followed standard checks of reporting and modelling [46–48] for this purpose. The outline of the disease pathway structure was reviewed by experienced modellers to avoid bias and checked by clinical experts to certify it was medically consistent.

## Results

Results of the primary Markov model analysis are presented as cost per QALYs gained, for a three-month cycle and over a time horizon of five years. Additionally, we also analysed the

**Table 6. Cost-effectiveness results for FlowOx™ therapy (one dose per annum) compared to standard care over one-year.**

| Scenario* | Cost (£) | QALYs | Incremental | | ICER |
| | | | Cost (£) | QALYs | Cost per QALY (£) |
|---|---|---|---|---|---|
| S0 | 2517.62 | 0.66 | -960.23 | 0.06 | Dominant |
| S1 | 1345.43 | 0.66 | -2132.41 | 0.06 | Dominant |
| S2 | 4792.38 | 0.66 | 1314.54 | 0.06 | 20672.61 |
| SC | 3477.84 | 0.60 | | | NA |

*S0 = FlowOx™+nominal care / S1 = FlowOx™ Only / S2 = FlowOx™+SC / SC = Standard Care.

effect of dosage and different intervention combinations, as described in the methods section. Please note incremental cost-effectiveness ratio (ICER) estimates will vary due to rounding.

The results from the modelled 1000 Monte Carlo simulations indicate that FlowOx™ therapy, regardless of whether standard, nominal or no additional care was provided, was a cost-effective treatment within this hypothetical cohort, based on a single three month dose of the intervention per year (see Tables 6 and 7). The probabilistic sensitivity analysis of this cost-effectiveness model gave a robust procedure for the analysis of parameters for uncertainty.

Examining the base case results in more detail, for a single three month annual dose, patients in scenario S0 (FlowOx™ therapy plus nominal care) incurred a total annual NHS cost of £2,518 and experienced an average of 0.66 QALYs, while patients receiving standard care incurred a total annual NHS cost of £3,478 and experienced 0.60 QALYs. This produces a cost-effective and cost saving ICER with a dominant cost per QALY, in favour of FlowOx™ therapy (see Table 6). When extrapolated to a five year time horizon, the intervention remains cost saving, cost-effective and dominant (see Table 7). The cost incurred from the intervention showed that S1 scenario always had the least cost while the S2 scenario had the most cost (see Tables 6 and 7).

For the base case FlowOx™ plus nominal care (S0) intervention, all points are distributed in the South-East quadrant signifying greater effect and less cost, in comparison to standard care. Conversely, for the FlowOx™ plus standard care (S2) intervention, all points are located in the North-East quadrant signifying greater effect and greater cost, in comparison to standard care. The distribution of the Monte Carlo simulation is shown in the ICER plane scatter plot (Fig 3).

A subgroup analysis of the cohort by age group is presented in Table 8, with effect discount rates varied for all intervention scenarios, in comparison with standard care. A similar pattern is observed in the FlowOx™ therapy plus nominal care (S0) and the FlowOx™ only (S1) scenarios; both scenarios appeared to be more effective and less expensive than standard care, and hence produce cost-effective and cost saving outcomes. The result was different for FlowOx™ therapy plus standard care (S2), which was more expensive than standard care but also more effective.

**Table 7. Cost-effectiveness results for FlowOx™ therapy (one dose per annum) compared to standard care over five years.**

| Scenario* | Cost (£) | QALYs | Incremental | | ICER |
| | | | Cost (£) | QALYs | Cost per QALY (£) |
|---|---|---|---|---|---|
| S0 | 12704.16 | 2.53 | -2819.52 | 0.27 | Dominant |
| S1 | 5574.55 | 2.53 | -9949.13 | 0.27 | Dominant |
| S2 | 20440.37 | 2.53 | 4916.70 | 0.27 | 18553.54 |
| SC | 15523.54 | 2.26 | | | NA |

*S0 = FlowOx™+nominal care / S1 = FlowOx™ Only / S2 = FlowOx™+SC / SC = Standard Care.

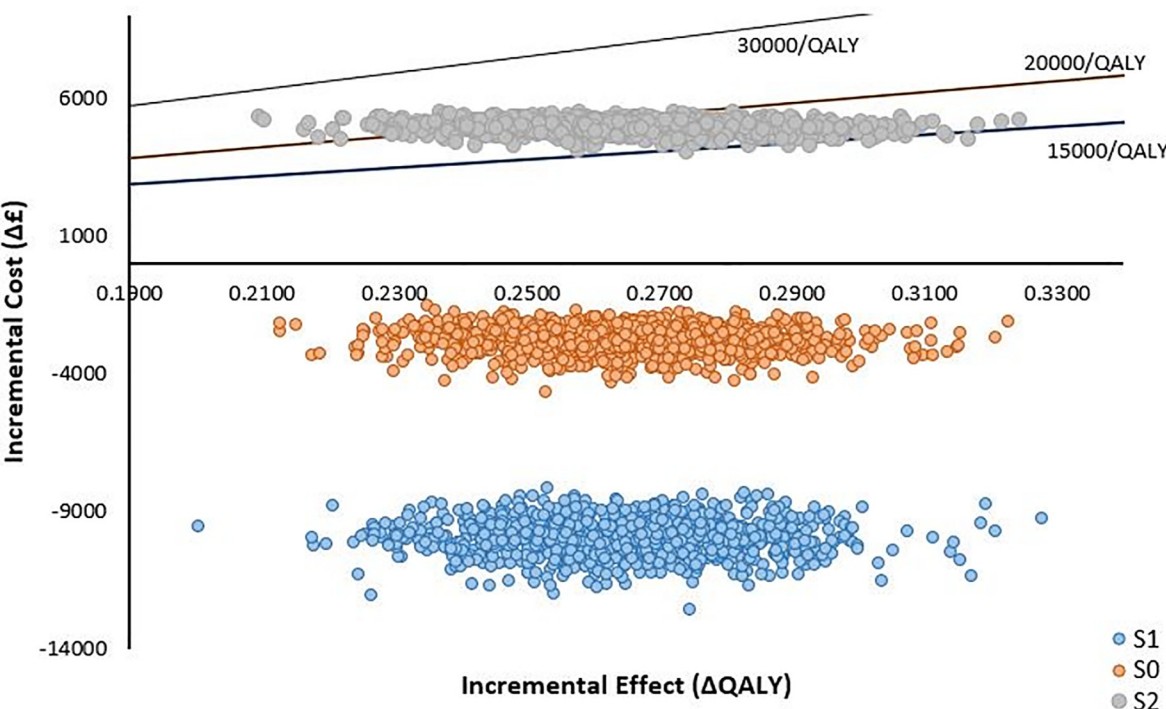

**Fig 3. ICER plane scatter plot showing the distribution of the ratio of incremental cost to effectiveness of FlowOx™ therapy plus standard care (S2) (one dose per annum).**

**Table 8. Summary cost and QALY outcomes for different age groups, intervention types and at different effect discount rates: Five year interval / one dose per annum.**

| Age | Scenario* | 1.5% Discount Effect | | 0% Discount Effect | | 3.5% Discount Effect | |
|---|---|---|---|---|---|---|---|
| | | Cost (£) | QALYs | Cost (£) | QALYs | Cost (£) | QALYs |
| 50–59 | S0 | 12704.16 | 2.77 | 12704.16 | 2.85 | 12704.16 | 2.66 |
| | S1 | 5574.55 | 2.77 | 5574.55 | 2.85 | 5574.55 | 2.66 |
| | S2 | 20440.37 | 2.77 | 20440.37 | 2.85 | 20440.37 | 2.66 |
| | SC | 15523.67 | 2.48 | 15523.67 | 2.55 | 15523.67 | 2.38 |
| 60–69 | S0 | 12704.16 | 2.77 | 12704.16 | 2.85 | 12704.16 | 2.66 |
| | S1 | 5574.55 | 2.77 | 5574.55 | 2.85 | 5574.55 | 2.66 |
| | S2 | 20440.37 | 2.77 | 20440.37 | 2.85 | 20440.37 | 2.66 |
| | SC | 15523.67 | 2.48 | 15523.67 | 2.55 | 15523.67 | 2.38 |
| 70–79 | S0 | 12704.16 | 2.66 | 12704.16 | 2.74 | 12704.16 | 2.56 |
| | S1 | 5574.55 | 2.66 | 5574.55 | 2.74 | 5574.55 | 2.56 |
| | S2 | 20440.37 | 2.66 | 20440.37 | 2.74 | 20440.37 | 2.56 |
| | SC | 15523.67 | 2.38 | 15523.67 | 2.45 | 15523.67 | 2.29 |
| ≥80 | S0 | 12704.16 | 2.63 | 12704.16 | 2.71 | 12704.16 | 2.53 |
| | S1 | 5574.55 | 2.63 | 5574.55 | 2.71 | 5574.55 | 2.53 |
| | S2 | 20440.37 | 2.63 | 20440.37 | 2.71 | 20440.37 | 2.53 |
| | SC | 15523.67 | 2.35 | 15523.67 | 2.42 | 15523.67 | 2.26 |

*S0 = FlowOx™+nominal care / S1 = FlowOx™ Only / S2 = FlowOx™+SC / SC = Standard Care.

**Table 9. Probability of cost-effectiveness at various WTP thresholds: One dose per annum.**

| Age | Intervention | One year WTP (£) | | | Five year WTP (£) | | |
|---|---|---|---|---|---|---|---|
| | | 15,000 | 20,000 | 30,000 | 15,000 | 20,000 | 30,000 |
| 50–59 | S2 vs SC | 0.000 | 0.566 | 1.000 | 0.026 | 0.940 | 1.000 |
| 60–69 | S2 vs SC | 0.000 | 0.597 | 1.000 | 0.016 | 0.924 | 1.000 |
| 70–79 | S2 vs SC | 0.000 | 0.353 | 1.000 | 0.017 | 0.829 | 1.000 |
| ≥80 | S2 vs SC | 0.000 | 0.286 | 1.000 | 0.012 | 0.786 | 1.000 |

*S2 = FlowOx™+SC / SC = Standard Care.

The WTP threshold in the UK is typically considered to be £20,000-£30,000 per QALY gained. The results of the Monte Carlo simulations for various WTP thresholds and various age groups are shown in Table 9 for the highest cost scenario of FlowOx™ therapy plus standard care (S2). In all scenarios the intervention appears to have a high probability of cost-effectiveness, compared to standard care, when provided as a single annual dose.

Irrespective of the age distribution, the results show that at a WTP threshold of £15,000 per QALY gained the probability of FlowOx™ therapy in addition to standard care being cost-effective is highly unlikely in this cohort, but at £30,000 per QALY gained the probability of cost-effectiveness is much higher. Linking the cost-effectiveness to probability accounts for uncertainty in the intervention outcomes. This is well represented in the cost-effectiveness acceptability curve (CEAC) (Fig 4).

IMNB was assessed to further establish the case for the robustness of the Markov model with limited uncertainty, the 95% confidence interval of the CEAC curve with a threshold of

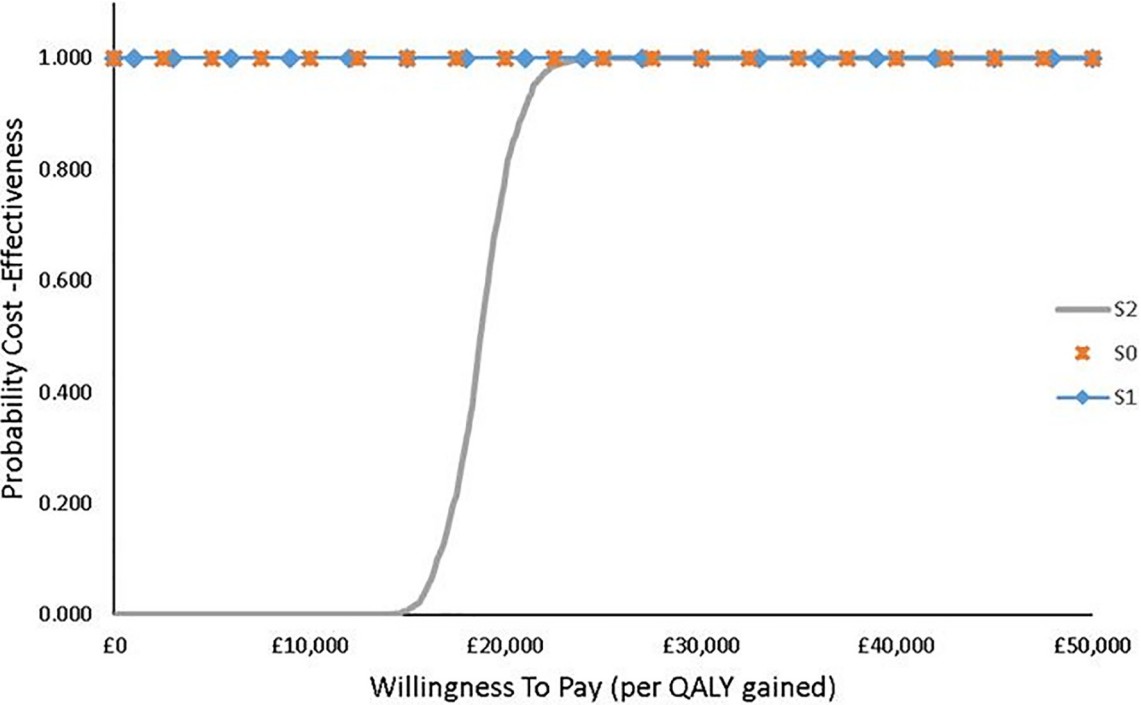

**Fig 4. CEAC showing the probability of cost-effectiveness of FlowOx™ therapy plus standard care at different WTP threshold levels (all ages).**

**Table 10. Probabilities and confidence intervals of cost-effectiveness at different WTP thresholds (all ages).**

| WTP Threshold | Probability of cost-effectiveness (CI) |
|---|---|
| £15,000 | FlowOx™ = 0.006 (0.000–0.001) |
| £20,000 | FlowOx™ = 0.800 (0.052–0.105) |
| £30,000 | FlowOx™ = 1.000 |

£50000 has a lower limit probability of 0.59 and an upper limit of 0.67. The CIs at given WTP thresholds are given in Table 10.

In order to examine the impact of dose on cost-effectiveness estimates, we conducted a continuous daily use sensitivity analysis, based on a hypothetical situation where FlowOx™ therapy does not produce a one year improving effect from a single three month dose and is therefore used continuously. In all scenarios the cost per QALY estimates are significantly higher, with only FlowOx™ therapy alone (S1) falling below the upper cost per QALY WTP threshold of £30,000 (see Table 11).

Looking at the base case results for this sensitivity analysis in more detail, the findings show that the cost of continuous FlowOx™ therapy plus nominal care (S0) would be more expensive than standard care in all age groups but would also be more effective. However, the cost per QALY estimates are high, ranging from £47,112 to £49,388 (see Table 12).

## Discussion

This economic evaluation investigates the cost-effectiveness of novel FlowOx™ therapy with nominal care using a Markov Monte Carlo simulation modelling technique. The base case result gave a cost-effective and cost saving ICER with dominant cost per QALY gained in comparison to the standard care. In the continuous case plus nominal care analysis, the ICER values were not cost-effective as they were above the UK willingness to pay threshold of £20,000 to £30,000. Sensitivity analysis showed that FlowOx™ therapy was cost-effective for a rental price to the NHS at £15 per person per day, for a cohort of patients with symptomatic PAD ranging from mild to severe. Similar to the results obtained in this study, other studies [49,50] support positive outcomes of clinical effects on granulation tissue formation with intermittent negative therapy.

### Summary of findings

From an NHS perspective, our base case analysis of FlowOx™ therapy plus nominal care (S0), delivered as a single annual dose, was cost-effective in terms of QALYs gained and cost saving at a payer threshold of £20,000 to 30,000 per QALY, when compared to standard care. Potential cost-savings are likely to come from substitution of FlowOx™ therapy for ongoing direct

**Table 11. Cost-effectiveness results for FlowOx™ therapy (continuous treatment) compared to standard care.**

| Scenario* | One year treatment | | | Five years treatment | | |
|---|---|---|---|---|---|---|
| | Cost | QALY | Cost per QALY | Cost | QALY | Cost per QALY |
| S0 | 6453.93 | 0.66 | 46802.09 | 28611.37 | 2.53 | 49387.46 |
| S1 | 5281.74 | 0.66 | 28368.21 | 21481.77 | 2.53 | 22483.33 |
| S2 | 8728.70 | 0.66 | 82575.26 | 36347.59 | 2.53 | 78580.68 |
| SC | 3477.84 | 0.60 | | 15523.67 | 2.26 | |

*S0 = FlowOx™+nominal care / S1 = FlowOx™ Only / S2 = FlowOx™+SC / SC = Standard Care.

**Table 12. Cost-effectiveness results for continuous FlowOx™ therapy plus nominal care compared to standard care.**

| Age | Scenario* | Cost (£) | QALYs | Incremental | | ICER |
| | | | | Cost (£) | QALYs | Cost per QALY (£) |
|---|---|---|---|---|---|---|
| 50–59 | S0 | 28611.37 | 2.66 | | | |
| | SC | 15523.67 | 2.38 | 13087.70 | 0.28 | 47111.66 |
| 60–69 | S0 | 28611.37 | 2.66 | | | |
| | SC | 15523.67 | 2.38 | 13087.70 | 0.28 | 47111.66 |
| 70–79 | S0 | 28611.37 | 2.56 | | | |
| | SC | 15523.67 | 2.29 | 13087.70 | 0.27 | 48829.30 |
| ≥80 | S0 | 28611.37 | 2.53 | | | |
| | SC | 15523.67 | 2.26 | 13087.70 | 0.27 | 49387.46 |

*S0 = FlowOx™+nominal care / SC = Standard Care.

care from nurses, health visitors and other healthcare professionals. The results from the base case analysis were robust following the conduct of deterministic and probabilistic sensitivity analysis. Interestingly, continuous FlowOx™ therapy does not appear to be cost-effective, due to significantly increased cost (assuming health outcomes are not significantly better than the single dose intervention).

## Strengths of our model

We believe that the data used in this economic model, drawn from Stage 2 trials of FlowOx™ therapy, is relevant to health service commissioners in the UK and internationally. The dataset studied for the relative risk and benefit covers most stages of disease state involved in PAD. We assessed the external validity of our model by taking it back to the clinicians and comparing it with existing literature.

## Limitations of our model

The sample size did not permit for analysis within disease severity groups. This would have informed the analysis on which disease state had the most benefit from FlowOx™ therapy. As such, we are unable to determine the optimum time point to initiate FlowOx™ therapy in order to maximise cost-effectiveness. Transition probabilities would have been best described from the study data, as this study is unique and imported estimates from other studies had to be adjusted to fit the model.

We chose a three month cycle for our model to reflect the primary data sources, which had limited follow-up periods. A follow-up of one year would have generated reliable evidence of the impact of one dosage per year, however this data was not available. Due to the need to make assumptions and draw data from various sources, the five year time horizon may not be a true reflection of reality. For instance, the impact of adherence and compliance are not considered in the model. Likewise, we were not able to use the primary data to determine the clinical benefits of continuous treatment versus a single annual dose, thus analysing the impact of dosage was predominantly focussed on hypothetical increases in cost.

Given these issues, we did not feel that extrapolating the data further, for instance in a lifetime analysis was not necessary or appropriate. Additional robust data is required to draw further conclusions about the cost-effectiveness of FlowOx™ therapy over a lifetime analysis. The costing of the FlowOx™ therapy within this model should be viewed with caution, as the rental cost appears to be quite low and may vary over time as more information on its value and effectiveness becomes available. In reality, this cost may be higher, and thus more real-world

cost analysis is needed to understand the true financial implications of FlowOx™ therapy. Due to the limited data available to us, we were not able to develop a micro-costing of FlowOx™ therapy.

### Unanswered questions

A qualitative study of patients using FlowOx™ therapy might be useful to understand the lived experience of the process of care and patient preferences around place/substitution of care. During the LLIFT pilot trial it was noted that compliance and adherence were significant issues, thus more research is needed on how this will impact cost-effectiveness in the long-term.

### Implications for commissioning

Provision of FlowOx™ therapy for use in the patients' own homes would require an upfront investment to rent or purchase sufficient FlowOx™ devices. There would be transport and regular maintenance costs associated with incorporation of FlowOx™ therapy into the care pathway, and at present it is unclear who would be responsible for paying for these costs.

### Meaning of the study

Adding FlowOx™ therapy to the patient pathway for patients with mild to severe PAD may be cost-effective to the NHS at a cost per QALY threshold of £20,000–30,000. However, this is likely to be dependent on disease severity, prognosis, comorbidities, dosage, adherence/compliance and wider costs associated with FlowOx™ therapy. Although the results from this model are promising, they are not conclusive.

### Conclusions

FlowOx™ therapy in combination with tailored nominal care appropriate for the stage of PAD appears to improve the health-related QoL of patients when compared with standard care, and may also be cost saving for the NHS. The results therefore indicate that FlowOx™ therapy may be an economically viable use of the NHS resources, but this is likely to depend on dosage, disease severity and the extent to which FlowOx™ therapy can be used to substitute direct care. The results obtained in this study show positive outcomes similar to the FlowOx^TM therapy trial conducted in Norway [51,52] which showed increase blood flow and wound healing as a result of FlowOx^TM therapy. Further research is needed to measure the impact of FlowOx™ therapy on resource use, in order to provide a definitive economic evaluation. A budget impact analysis assessing the likely costs of roll out within the NHS would help inform policymakers.

### Supporting information

**S1 Appendix. Model parameters.**
(DOCX)

### Acknowledgments

We would like to thank our colleagues Dr Jo Charles and Dr Lorna Tuersley, who helped to collate unit costs for this study and Naa Amua Quaye who helped with formatting a draft of the manuscript.

## Author Contributions

**Conceptualization:** Nathan Bray, Henrik Hoel.

**Data curation:** Victory 'Segun Ezeofor, Farina Hashmi, Daniel Parker.

**Formal analysis:** Victory 'Segun Ezeofor.

**Funding acquisition:** Nathan Bray, Henrik Hoel.

**Investigation:** Henrik Hoel, Daniel Parker.

**Methodology:** Victory 'Segun Ezeofor, Nathan Bray.

**Project administration:** Henrik Hoel.

**Software:** Victory 'Segun Ezeofor.

**Supervision:** Nathan Bray, Henrik Hoel, Rhiannon Tudor Edwards.

**Validation:** Victory 'Segun Ezeofor.

**Visualization:** Nathan Bray.

**Writing – original draft:** Victory 'Segun Ezeofor, Nathan Bray, Lucy Bryning, Rhiannon Tudor Edwards.

**Writing – review & editing:** Victory 'Segun Ezeofor, Nathan Bray, Lucy Bryning, Farina Hashmi, Henrik Hoel, Daniel Parker, Rhiannon Tudor Edwards.

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
