## [Decision Letter · Decision Letter 0]

7 Aug 2020

PONE-D-20-16557

Economic model to examine the cost-effectiveness of FlowOx™ home therapy compared to standard care in patients with peripheral artery disease

PLOS ONE

Dear Dr. Ezeofor,

Thank you for submitting your manuscript to PLOS ONE. After careful consideration, we feel that it has merit but does not fully meet PLOS ONE’s publication criteria as it currently stands. Therefore, we invite you to submit a revised version of the manuscript that addresses the points raised during the review process.

Please do address the Review comments on presentation of results, avoiding the primary presentation of results using negative cost-effectiveness ratios.  It would be helpful to clarify the statement made on the cost-effectiveness of the FlowOx in addition to standard care (as Review comments), and to reconsider the appropriateness of the discount rates applied.

Whilst I note one of the Reviewers has requested an update to the unit costs (year of cost data), I would be happy for you to retain the current approach if the requested changed involved excessive additional work.  However, if you could address that request it would be helpful.

Please do ensure any potential conflicts of interest are clearly stated, given at least one of the co-authors has an affiliation to Otivio AS.

We look forward to receiving your revised manuscript.

Kind regards,

Colin Green, Ph.D

Academic Editor

PLOS ONE

Journal Requirements:

2. Thank you for including your competing interests statement;

"HH is employed by Otivio AS with funding from The Research Council of Norway (grant no. 285758). Otivio AS has the commercial rights to the INP technology used in this study. The authors alone are responsible for the content and writing of the paper."

We note that one or more of the authors are employed by a commercial company:Otivio AS

Reviewers' comments:

Reviewer's Responses to Questions

**Comments to the Author**

1. Is the manuscript technically sound, and do the data support the conclusions?

Reviewer #1: Yes

Reviewer #2: Partly

2. Has the statistical analysis been performed appropriately and rigorously? 

Reviewer #1: Yes

Reviewer #2: Yes

3. Have the authors made all data underlying the findings in their manuscript fully available?

Reviewer #1: Yes

Reviewer #2: Yes

4. Is the manuscript presented in an intelligible fashion and written in standard English?

Reviewer #1: Yes

Reviewer #2: Yes

5. Review Comments to the Author

Reviewer #1: A very nice analysis has been presented exploring the potential cost-effectiveness of FlowOx therapy compared to standard care. Congratulations on the submission. I have a few minor comments, which are listed below:

- Introduction: A nice background to the condition has been presented, and the financial burden of progressing to end-stage PAD is clear. However, upon presentation of the intervention, I would like you to clearly state at what point in the treatment pathway it is likely to be used, i.e. pre-analgesics/post analgesics/prior to significant progression of the condition and surgical intervention. This would allow me to put use of the intervention in the overall context of progression of the disease right at the beginning of the paper.

- Methods: Moloney et al. carried out a systematic review of economic models used to compare methods of diagnosing PAD, which summarises evidence in this area (Systematic Review of Economic Models Used to Compare Techniques for Detecting Peripheral Arterial Disease). Was this identified in your searches and could any of the models outlined here have been used to inform your own analysis?

As mentioned previously, you talk about use of the intervention for the purpose of treating PAD patients in your 'Model overview' and 'Decision model' sections, but I am still unclear at this point as to the stage in the treatment pathway at which the intervention would be introduced, beyond knowing it will cover mild/severe stage PAD.

Is 5 years a sufficient time horizon to capture all relevant costs and effects? You state in the introduction that the condition is associated with prolonged morbidity. If that's the case and the intervention can potentially reduce associated morbidity, then I would suggest that a longer time horizon should be used. At the very least, if it can be definitively argued that 5 years is a sufficient time horizon, it may still be worth carrying out a lifetime analysis as part of your sensitivity analysis to explore any change in results, even if there is uncertainty surrounding the long-term extrapolation.

I suggest you update your costs to a 2018/19 price year given that we are into 2020 now.

In your 'Sources of cost data' section I would like to see more detail as to what is actually included in each health state cost, beyond 'community, inpatient and outpatient' costs.

When you state that 'expert opinion' was used, please clearly indicate the source of this opinion.

- Results: I note that you have presented negative ICERs, which I suggest shouldn't be done. Instead, present these results in terms of 'dominant', 'dominated' etc.

In your cost-effectiveness plane, please present the WTP threshold in order to help interpret the scatter-plot.

In your CEAC, make it clear that the WTP values are £ sterling, and reduce your prob. cost-effective values to two decimal places.

- Discussion: In your discussion and concluding sections, I would like to see some reference to other similar studies that have been conducted, if at all, to help put the results of your own analysis in context.

Reviewer #2: I would like to thank the authors for giving me the opportunity to review their article. As they will see most of my comments relate to how the results are presented and some of the assumptions (e.g. discount rates) underpinning their model.

Abstract

1) What is nominal care? Is it the same as standard care?

2) In the results, the authors report mean treatment costs. Are these just the costs of the actual treatment, or do these include follow-up healthcare costs. If so, please amend the terminology.

3) The authors report a negative ICER. Although, not incorrect, it might lead to potential confusion, as a negative ICER could be arrived at by different combinations of negative incremental costs or outcomes. Would it be best if the authors reported that FlowOx therapy was dominant?

4) I am very confused by the following: “FlowOx™ therapy in addition to standard care was the least cost-effective scenario at £17,823 per QALY gained.” As opposed to what? In addition, if FlowOx + standard care yielded more QALYs than FlowOx therapy on its own, would FlowOx + SC not be recommended?

5) I missed any probabilistic sensitivity results in the abstract.

Methods

6) Source needed for the assertion: Cost data associated with PAD have a wide 237 variance, which has a direct impact on average cost estimates.”

7) More details are required on which costs were derived from the literature and which ones were derived from expert opinion.

8) I am very dubious at the differential discounting rate applied by the authors. The UK’s current recommendation is: “a rate of 3.5% for costs and effects”. As far as I am aware, the only exception to this is: “for therapies with long-term benefits, where differential discounting is recommended with a lower discount rate for health effects of 1.5%.”

Results

9) Although I can understand why the authors compare all interventions to SC, interventions should also be compared against each other. Therefore, at one year for e.g., S0 should be compared to S1, SC to S0 and S2 to SC (before removing interventions that are dominated)

10) QALYs should be rounded to the 3rd or 4th decimal point. As results are presented, it implies there is absolutely no difference in QALY gain between the 3 interventions.

11) I found Tables 7 and 8 absolutely redundant as they present all the information in Table 6.

12) Why are CEACs not presented including all four of the interventions assessed?

13) I also missed that no break-up of costs was given. How much of the total costs are due to: FlowOx therapy, amputations, early stages of disease etc…

6. PLOS authors have the option to publish the peer review history of their article (what does this mean?). If published, this will include your full peer review and any attached files.

Reviewer #1: No

Reviewer #2: **Yes: **Ramon Luengo-Fernandez

---

## [Author Response · Author response to Decision Letter 0]

2 Dec 2020

Review 1

- Introduction: A nice background to the condition has been presented, and the financial burden of progressing to end-stage PAD is clear. However, upon presentation of the intervention, I would like you to clearly state at what point in the treatment pathway it is likely to be used, i.e. pre-analgesics/post analgesics/prior to significant progression of the condition and surgical intervention. This would allow me to put use of the intervention in the overall context of progression of the disease right at the beginning of the paper.

Ans- We have included statements to answer this query in the introduction and decision model section (see Page 6-7, Lines 121-126 and Pages 8-9, Lines 169-173).

- As mentioned previously, you talk about use of the intervention for the purpose of treating PAD patients in your 'Model overview' and 'Decision model' sections, but I am still unclear at this point as to the stage in the treatment pathway at which the intervention would be introduced, beyond knowing it will cover mild/severe stage PAD.

Ans- We have clarified this in the manuscript (see Page 6-7, Lines 121-126 and Pages 8-9, Lines 169-173) and have also included references to a clinical trial of the FlowOXTM device from another study in the introduction section (Page 6, Lines 116-118).

- Methods: Moloney et al. carried out a systematic review of economic models used to compare methods of diagnosing PAD, which summarises evidence in this area (Systematic Review of Economic Models Used to Compare Techniques for Detecting Peripheral Arterial Disease). Was this identified in your searches and could any of the models outlined here have been used to inform your own analysis?

Ans- This is a very nice systematic review, though we did not use it in this paper we have pencilled it down for the main trial of the FlowOXTM therapy.

-Is 5 years a sufficient time horizon to capture all relevant costs and effects? You state in the introduction that the condition is associated with prolonged morbidity. If that's the case and the intervention can potentially reduce associated morbidity, then I would suggest that a longer time horizon should be used. At the very least, if it can be definitively argued that 5 years is a sufficient time horizon, it may still be worth carrying out a lifetime analysis as part of your sensitivity analysis to explore any change in results, even if there is uncertainty surrounding the long-term extrapolation.

Ans- This would be investigated in larger scale trial we are looking into conducting in the future. This study was more like a feasibility trial and with a lot of missing data we have decided not to conduct a lifetime analysis. We feel that it would be a step beyond the available data to extrapolate the findings to a lifetime analysis. We have added a caveat to the discussion to explain our position: “Given these issues, we did not feel that extrapolating the data further, for instance in a lifetime analysis was necessary. Additional robust data is required to draw further conclusions about the cost-effectiveness of FlowOx™ therapy over a lifetime analysis” (see page 27, lines 441-443)

-I suggest you update your costs to a 2018/19 price year given that we are into 2020 now.

Ans- Yes, agreed. All cost in the study where necessary were inflated to 2018/19 in the computation (lines 258-259).

 - In your 'Sources of cost data' section I would like to see more detail as to what is actually included in each health state cost, beyond 'community, inpatient and outpatient' costs.

Ans- In this study cost were obtained from literature and as noted these costs for PAD have some variance hence expert opinion were used as guidance in choosing which costs and reference were suitable for this study. Tables A, B and D in the appendix gives the breakdown to Table 3 with details to cost applied in the study and the references.

- When you state that 'expert opinion' was used, please clearly indicate the source of this opinion.

Ans- Expert advice were essential in selecting which cost were necessary to this study from a wide range of cost (see Source of Cost section, Page 14-15, Lines 262 - 264).

- Results: I note that you have presented negative ICERs, which I suggest shouldn't be done. Instead, present these results in terms of 'dominant', 'dominated' etc.

Ans- Agreed and have been changed (Tables 6-7).

- In your cost-effectiveness plane, please present the WTP threshold in order to help interpret the scatter-plot.

Ans- This has been added to the scatter plot as suggested (Figure 3). 

- In your CEAC, make it clear that the WTP values are £ sterling, and reduce your prob. cost-effective values to two decimal places.

Ans- Agreed and this has been amended (Figure 4).

- Discussion: In your discussion and concluding sections, I would like to see some reference to other similar studies that have been conducted, if at all, to help put the results of your own analysis in context.

Ans- Agreed and have been added in the Discussion and conclusion section (see Page 24-25, Lines 408-410; Pages 27, Lines 471 -473).

Review 2

1) What is nominal care? Is it the same as standard care?

Ans- Nominal care is a reduced standard care, such that there are fewer visit by the nurse and medication is reduced as shown in Table A of the Appendix, while standard care is seen in tables A and D (see Decision model section, Page 9, Lines 181-184).

2) In the results, the authors report mean treatment costs. Are these just the costs of the actual treatment, or do these include follow-up healthcare costs. If so, please amend the terminology.

Ans- Agreed and Amended (lines 34-36).

3) The authors report a negative ICER. Although, not incorrect, it might lead to potential confusion, as a negative ICER could be arrived at by different combinations of negative incremental costs or outcomes. Would it be best if the authors reported that FlowOx therapy was dominant?

Ans- Yes, Agreed and Amended (Tables 6-7).

4) I am very confused by the following: “FlowOx™ therapy in addition to standard care was the least cost-effective scenario at £17,823 per QALY gained.” As opposed to what? In addition, if FlowOx + standard care yielded more QALYs than FlowOx therapy on its own, would FlowOx + SC not be recommended?

Ans- This statement has been taken out

5) I missed any probabilistic sensitivity results in the abstract.

Ans- A probabilistic sensitivity result has been included in the abstract (see Abstract, Page 2, Lines 38-39).

6) Source needed for the assertion: Cost data associated with PAD have a wide 237 variance, which has a direct impact on average cost estimates.

Ans- We have included cost data in Tables A, B and D of the appendix to show the breakdown of all cost applied in this analysis.

7) More details are required on which costs were derived from the literature and which ones were derived from expert opinion.

Ans- Expert advice were essential in selecting which cost were necessary to this study from a wide range of cost (see Source of Cost section, Page 14-15, Lines 262 - 264).

In this study cost were obtained from literature and as noted these costs for PAD have some variance hence expert opinion were used as guidance in choosing which costs and reference were suitable for this study. Tables A, B and D in the appendix give details to cost applied in the study and the references. 

8) I am very dubious at the differential discounting rate applied by the authors. The UK’s current recommendation is: “a rate of 3.5% for costs and effects”. As far as I am aware, the only exception to this is: “for therapies with long-term benefits, where differential discounting is recommended with a lower discount rate for health effects of 1.5%.”

Ans- A rate of 3.5% has now been implemented on the Effect in accordance with the NICE guideline [43] (See Discounting section, Page 15, Line 281 - 284). 

9) Although I can understand why the authors compare all interventions to SC, interventions should also be compared against each other. Therefore, at one year for e.g., S0 should be compared to S1, SC to S0 and S2 to SC (before removing interventions that are dominated)

Ans- Data for the EQ-5D showed very similar results as shown in Tables 6 and 7, with this the change in effect will be zero and thus it is not particularly useful to report these findings. This will be one of the points we will focus on during the larger scale trial.

10) QALYs should be rounded to the 3rd or 4th decimal point. As results are presented, it implies there is absolutely no difference in QALY gain between the 3 interventions.

Ans- All QALYs have been rounded to 2dp as requested (Tables 6 – 8).

11) I found Tables 7 and 8 absolutely redundant as they present all the information in Table 6.

Ans- Table 7 has been changed and table 8 removed.

12) Why are CEACs not presented including all four of the interventions assessed?

Ans- These have now been included (Figure 4).

13) I also missed that no break-up of costs was given. How much of the total costs are due to: FlowOx therapy, amputations, early stages of disease etc…

Ans- In this study cost were obtained from literature and as noted these costs for PAD have some variance hence expert opinion were used as guidance in choosing which costs and reference were suitable for this study. The breakdown of the cost are presented in Tables A, B and D in the appendix.

---

## [Decision Letter · Decision Letter 1]

18 Dec 2020

Economic model to examine the cost-effectiveness of FlowOx™ home therapy compared to standard care in patients with peripheral artery disease

PONE-D-20-16557R1

Dear Dr. Ezeofor,

We’re pleased to inform you that your manuscript has been judged scientifically suitable for publication and will be formally accepted for publication once it meets all outstanding technical requirements.

Kind regards,

Prof. Raffaele Serra, M.D., Ph.D

Academic Editor

PLOS ONE

Additional Editor Comments (optional):

amended manuscript is acceptable

Reviewers' comments:

Reviewer's Responses to Questions

**Comments to the Author**

1. If the authors have adequately addressed your comments raised in a previous round of review and you feel that this manuscript is now acceptable for publication, you may indicate that here to bypass the “Comments to the Author” section, enter your conflict of interest statement in the “Confidential to Editor” section, and submit your "Accept" recommendation.

Reviewer #1: All comments have been addressed

Reviewer #2: All comments have been addressed

2. Is the manuscript technically sound, and do the data support the conclusions?

Reviewer #1: Yes

Reviewer #2: Yes

3. Has the statistical analysis been performed appropriately and rigorously? 

Reviewer #1: Yes

Reviewer #2: Yes

4. Have the authors made all data underlying the findings in their manuscript fully available?

Reviewer #1: Yes

Reviewer #2: Yes

5. Is the manuscript presented in an intelligible fashion and written in standard English?

Reviewer #1: Yes

Reviewer #2: Yes

6. Review Comments to the Author

Reviewer #1: (No Response)

Reviewer #2: I would like to thank the authors for addressing my comments satisfactorily.

I would, however, would like to see QALYs rounded up to the fourth-fifth decimal point not just two (currently showing that three interventions are identical outcomes).

7. PLOS authors have the option to publish the peer review history of their article (what does this mean?). If published, this will include your full peer review and any attached files.

Reviewer #1: **Yes: **Eoin Moloney

Reviewer #2: **Yes: **Ramon Luengo-Fernandez

---

## [Editor Report · Acceptance letter]

28 Dec 2020

PONE-D-20-16557R1 

Economic model to examine the cost-effectiveness of FlowOx home therapy compared to standard care in patients with peripheral artery disease 

Dear Dr. Ezeofor:

I'm pleased to inform you that your manuscript has been deemed suitable for publication in PLOS ONE. Congratulations! Your manuscript is now with our production department. 

Kind regards, 

on behalf of

Prof. Raffaele Serra 

Academic Editor

PLOS ONE